# Simultaneous Analysis of D,L-Amino Acids in Human Urine Using a Chirality-Switchable Biaryl Axial Tag and Liquid Chromatography Electrospray Ionization Tandem Mass Spectrometry

**Masashi Harada [1]** [ID]**, Sachise Karakawa [1], Hiroshi Miyano [2] and Kazutaka Shimbo [1,\***

[1]   Research Institute for Bioscience Products & Fine Chemicals, Ajinomoto Co., Inc., 1-1 Suzuki-cho, Kawasaki-ku, Kawasaki-shi 210-8681, Japan; masashi_harada@ajinomoto.com (M.H.); sachise_karakawa@ajinomoto.com (S.K.)

[2]   R&D Planning Department, Ajinomoto Co., Inc., 1-15-1 Kyobashi, Chūō-ku, Tokyo 104-8315, Japan; hiroshi_miyano@ajinomoto.com

**\***   Correspondence: kazutaka_shinbo@ajinomoto.com; Tel.: +81-44-210-5832

**Abstract:** Although D,L-amino acids are symmetrical molecules, L-isomers are generally dominant in living organisms. However, it has been found that some D-amino acids also have biological functions. A new method for simultaneously analyzing D,L-amino acids in biological samples is required to allow unknown functions of D-amino acids to be investigated. D-Amino acids in urine are currently receiving increasing amounts of attention, particularly for screening for chronic kidney diseases. However, simultaneously analyzing D,L-amino acids in human urine is challenging because of interfering unknown compounds in urine. In this study, the axially chiral derivatizing agent (*R*)-4-nitrophenyl-*N*-[2-(diethylamino)-6,6-dimethyl-[1,1-biphenyl]-2-yl] carbamate hydrochloride was used to allow enantiomers of amino acids in human urine to be simultaneously determined by liquid chromatography electrospray ionization tandem mass spectrometry. The optimized method gave good linearities, precision results, and recoveries for 18 proteinogenic amino acids and their enantiomers and glycine. The chiral-switching method using (*S*)-4-nitrophenyl-*N*-[2-(diethylamino)-6, 6-dimethyl-[1,1-biphenyl]-2-yl]carbamate hydrochloride confirmed the expected concentrations of 32 of the 37 analytes. The method was successfully used to determine the concentrations of D-serine, D-alanine, D-asparagine, D-allothreonine, D-lysine, and the D-isomers of 10 other amino acids in five human volunteer urine samples.

**Keywords:** D-amino acid; chiral derivatization; axially chiral; urine analysis; liquid chromatography; mass spectrometry

## 1. Introduction

Amino acids play important roles in the body, as the building blocks of proteins and as intermediates in metabolic systems. Proteinogenic amino acids, except for glycine (Gly), have chirality. Although D,L-amino acids, a pair of enantiomers, are symmetrical molecules, stereospecific biological reactions accurately control the stereostructures of amino acids, and L-isomers of amino acids are dominant in natural systems. Therefore, D-isomers are minor enantiomers that have previously been thought not to have biological functions. However, in the last couple of decades, it has been found that D-amino acids do have biological functions. For example, D-serine (Ser) [1–4] and D-aspartate (Asp) [5–8] have biological functions in the brain and nervous system that have been investigated intensively. D-amino acids in blood have recently been investigated as potential biomarkers [9].

ᴅ-Amino acids have been found in mammal urine [10–13], and the concentrations of ᴅ-amino acids in urine have been used to screen for kidney diseases. Kidneys play important roles in controlling free ᴅ-amino acid concentrations in blood by excreting and metabolizing ᴅ-amino acids. Acute kidney injury in mice has been found to increase the ᴅ-Ser concentration in blood and decrease the ᴅ-Ser concentration in urine [14]. A relationship between chronic kidney disease and the concentrations of several ᴅ-amino acids in plasma has recently been found [15–17]. It has recently been suggested that ᴅ-alanine (Ala) in human urine has fundamental physiological functions related to resting–active conditions and could be a useful biomarker for sleeping–awake profiles [18]. In those studies, few ᴅ-amino acids in urine were analyzed or discussed. The strong relationships between kidney function and ᴅ-amino acid concentrations mean that further investigations using simultaneous ᴅ,ʟ-amino acid analyses are required to clarify the mechanisms involved and to identify appropriate biomarkers.

Advances in chiral analysis have allowed detailed studies of ᴅ-amino acid functions to be performed [19]. Chromatographic separation techniques such as high-performance liquid chromatography (HPLC) and gas chromatography have generally been used to separate ᴅ- and ʟ-amino acids. Chiral HPLC has most often been used for such analyses. Analyses have been performed using two- or three-dimensional HPLC with fluorescent derivatization [16,20,21], chiral HPLC of derivatized amino acids with mass spectrometry (MS) [9,22], and chiral HPLC of underivatized free amino acids with MS [23,24]. Two- or three-dimensional HPLC is selective and accurate but time-consuming and requires complex instrumentation. Derivatization followed by HPLC analysis with MS still requires more than 40 min per analysis for simultaneous determination, or only limited amino acids can be analyzed. Methods without derivatization cannot separate ᴅ,ʟ-proline (Pro), and are prone to matrix interferences, but can give fast analyses. Using chiral gas chromatography with harsh derivatization at a high temperature means some unstable amino acids have to be analyzed as the sums of amino acid concentrations (e.g., Asp as asparagine (Asn) plus Asp and glutamate (Glu) as glutamine (Gln) plus Glu), although good separation can be achieved.

Chiral derivatization methods have also been used in high-throughput methods for simultaneously determining amino acids. Various chiral derivatizing agents are available [25], and a method using (*R*)-4-nitrophenyl-*N*-[2-(diethylamino)-6,6-dimethyl-[1,1-biphenyl]-2-yl]carbamate hydrochloride (BiAC) has been found to separate all proteinogenic amino acid enantiomers in 14.5 min [26]. This derivatizing agent allows comprehensive, very sensitive, and very selective amino acid analysis to be performed, but has only been used to determine chiral amino acids in food samples up to now.

In this study, a method for simultaneously determining ᴅ,ʟ-amino acids in human urine using (*R*)-BiAC was developed. The conditions were modified and optimized for the analysis of biological samples. Urine generally contains large amounts of hydrophilic compounds [27] that can interfere with amino acid analysis by causing overlapping peaks. To ensure the concentrations obtained by the analysis using (*R*)-BiAC were accurate, the analytical results were verified using (*S*)-BiAC, the *S* isomer of the derivatization reagent.

## 2. Materials and Methods

### 2.1. Chemicals and Reagents

Amino acid standard mixture type H (a mixture of ʟ-Ala, ʟ-arginine (Arg), ʟ-Asp, ʟ-cystine, ʟ-glutamate (Glu), Gly, ʟ-histidine (His), ʟ-isoleucine (Ile), ʟ-leucine (Leu), ʟ-lysine (Lys), ʟ-methionine (Met), ʟ-phenylalanine (Phe), ʟ-Pro, ʟ-Ser, ʟ-threonine (Thr), ʟ-tyrosine (Tyr), and ʟ-valine (ʟ-Val), each at a concentration of 2.5 mM) was purchased from FUJIFILM Wako Pure Chemical (Osaka, Japan). ʟ-Asn, ʟ-Gln, ʟ-tryptophan (ʟ-Trp), ᴅ-Ala, ᴅ-Arg, ᴅ-Asn, ᴅ-Asp, ᴅ-Gln, ᴅ-Glu, ᴅ-His, ᴅ-Leu, ᴅ-Lys, ᴅ-Met, ᴅ-Phe, ᴅ-Pro, ᴅ-Ser, ᴅ-allothreonine (alloThr), ᴅ-Trp, ᴅ-Tyr, and ᴅ-Val were purchased from Sigma-Aldrich (St. Louis, MO, USA). ᴅ-Alloisoleucine (alloIle) was purchased from Tokyo Chemical Industry (Tokyo, Japan).



[$^{13}$C] and [$^{15}$N] uniformly labeled amino acids (ʟ-Ala, ʟ-Pro, and ʟ-Tyr) were purchased from Isotec (Tokyo, Japan). [$^{13}$C] and [$^{15}$N] uniformly labeled ʟ-Asn and ʟ-Asp were purchased from Spectra Stable Isotopes (Columbia, MD, USA). [$^{13}$C] and [$^{15}$N] uniformly labeled ʟ-Lys dihydrochloride, ʟ-Met, ʟ-Phe, ʟ-Ser, ʟ-Thr, ʟ-Val, and Gly were purchased from Cambridge Isotope Laboratories (Tewksbury, MA, USA). [$^{13}$C] and [$^{15}$N] uniformly labeled ʟ-Arg, ʟ-Gln, ʟ-Glu, ʟ-His, ʟ-Ile, ʟ-Leu, and ʟ-Trp were obtained from Ajinomoto (Tokyo, Japan).

The derivatizing agents (*R*)- and (*S*)-BiAC (Figure 1) were synthesized in our laboratory using a previously published method [26]. The precursors of the derivatizing agents were optically purified by chiral supercritical fluid chromatography and recycling HPLC. The derivatizing agents had optical purities of >99.9% ee for (*R*)-BiAC and 99.7% ee for (*S*)-BiAC.

**Figure 1.** Structures of the chirality-switchable biaryl axially chiral tags (**a**) (*R*)-4-nitrophenyl-*N*-[2-(diethylamino)-6,6-dimethyl-[1,1-biphenyl]-2-yl]carbamate hydrochloride (BiAC) and (**b**) (*S*)-BiAC.

Ammonium formate was purchased from Kanto Chemical Industry (Tokyo, Japan). Acetonitrile, sodium borate buffer (APDSTAG Wako borate buffer; 200 mM, pH 8.8), and formic acid were purchased from FUJIFILM Wako Pure Chemical. Water was purified using a Milli-Q gradient A10 system (Merck Millipore, Darmstadt, Germany).

### 2.2. Human Urine Samples

Fasting spot urine samples from five healthy volunteers were obtained following a protocol (16H-05) approved by the Institutional Review Board for human protection at the Institute for Innovation, Ajinomoto Co., Inc. and performed in accordance with the ethical principles stated in the Declaration of Helsinki, 1964.

### 2.3. Instrumentation

The ultra-HPLC system that was used was an Agilent 1290 Infinity II system (Agilent Technologies, Santa Clara, CA, USA) with a binary pump, degasser, autosampler, and column compartment. The HPLC system was coupled to a QTRAP 5500 triple quadrupole MS system (Sciex, Redwood City, CA, USA). The instruments were controlled and the data were processed using Sciex Analyst 1.6.2 and MultiQuant 3.0.2 software.

### 2.4. Standard Solution Preparation

Working standard solutions containing ᴅ- and ʟ-amino acids each at a concentration ratio of 1:10 were prepared by sequentially diluting the stock solutions with water. The amino acid concentrations (as ᴅ-amino acid concentration/ʟ-amino acid and Gly concentration, in µM) in the working standard solutions were 0.01/0.1 (STD-12), 0.025/0.25 (STD-11), 0.05/0.5 (STD-10), 0.1/1 (STD-9), 0.25/2.5 (STD-8), 0.5/5 (STD-7), 1/10 (STD-6), 2.5/25 (STD-5), 5/50 (STD-4), 10/100 (STD-3), 25/250 (STD-2), and 50/500 (STD-1). The ʟ-amino acids were ʟ-Ala, ʟ-Arg, ʟ-Asn, ʟ-Asp, ʟ-Gln, ʟ-Glu, Gly, ʟ-His, ʟ-Ile, ʟ-Leu, ʟ-Lys, ʟ-Met, ʟ-Pro, ʟ-Ser, ʟ-Thr, ʟ-Trp, ʟ-Tyr, ʟ-Val, ʟ-Phe, and ʟ-cystine (not analyzed). The ᴅ-amino acids were ᴅ-Ala, ᴅ-Arg, ᴅ-Asn, ᴅ-Asp, ᴅ-Gln, ᴅ-Glu, ᴅ-His, ᴅ-allolle, ᴅ-Leu, ᴅ-Lys, ᴅ-Met, ᴅ-Pro, ᴅ-Ser, ᴅ-alloThr, ᴅ-Trp, ᴅ-Tyr, ᴅ-Val, and ᴅ-Phe.

### 2.5. Internal Standard Solution Preparation

Isotope-labeled internal standard (IS) solutions (containing ᴅ- and ʟ-amino acids each at a ᴅ-/ʟ-amino acid concentration ratio of 1:10) were prepared using commercially available [$^{13}$C] and [$^{15}$N] uniformly labeled ʟ-amino acids and labeled ᴅ-amino acids prepared by racemization using previously published microwave heating conditions [26]. Several labeled unstable amino acids (e.g., Asn and Gln) were prepared using a patented racemization method [28]. A racemized amino acid solution was mixed with the corresponding labeled ʟ-amino acid solution in the proportions required to give a ᴅ-/ʟ-amino acid concentration ratio of 1:10.

### 2.6. Urine Sample Preparation and Derivatization

Each urine sample was collected in a 50 mL plastic tube and stored at −80 °C until analysis. A pooled urine sample was prepared by mixing the same volumes of multiple individual urine samples. A 50 μL aliquot of a pooled or individual urine sample was mixed with 450 μL of water. For a recovery test, 50 μL of pooled urine, 250 μL of a standard mixture, and 200 μL of water were mixed. A 20 μL of the sample or a standard mixture (to allow calibration curves to be drawn) was then mixed with 20 μL of the IS solution and 40 μL of acetonitrile. The mixture was centrifuged at 20,000× *g* for 10 min at 20 °C. A 20 μL aliquot was transferred to a 1.5 mL polypropylene micro test tube and then mixed with 60 μL of a 1:1 *v/v* mixture of 200 mmol/L sodium borate buffer and acetonitrile. A 20 μL aliquot of a freshly prepared solution containing 10 mg/mL (*R*)- or (*S*)-BiAC in acetonitrile was added, then the mixture was vigorously mixed and incubated at 55 °C for 10 min. A 200 μL aliquot of 0.1% aqueous formic acid was then added to quench the reaction.

### 2.7. LC-MS/MS Analysis

The derivatized amino acids were separated using a YMC Triart Phenyl column (1.9 μm particle size, 75 mm long, 2.1 mm i.d.; YMC, Kyoto, Japan) at 40 °C. Mobile phase A was 0.1% formic acid containing 10 mM ammonium formate in water, and mobile phase B was a 95:5 *v/v* mixture of acetonitrile and water. The mobile phase gradient program was: 0–3.0 min, 14% mobile phase B; 3.0–14.3 min, 14% to 33% mobile phase B; 14.3–17.1 min, 33% to 45% mobile phase B; 17.1–18.0 min, 90% mobile phase B; 18.0–18.1 min, 90% to 14% mobile phase B; and 18.1–20.0 min, 14% mobile phase B. The flow rate was 0.4 mL/min, and the injection volume was 5 μL. The MS parameters were: curtain gas setting 40; ion spray voltage 5500 V; temperature 600 °C; ion source gas 1 setting 70; ion source gas 2 setting 70; collision activated dissociation setting 8; and entrance potential setting 10. Quantitative analysis was performed in selected reaction monitoring mode. The precursor ions that were monitored and the optimized parameters for the BiAC-tagged amino acids (declustering potential, collision energy, and collision cell exit potential) are shown in Table A1 (Appendix A). Product ion *m/z* 295.1 was used for all of the analytes. The calibration curve for each amino acid was constructed by plotting the analyte/IS peak area ratio for the BiAC-tagged derivatives.

## 3. Results and Discussion

### 3.1. Optimization of the LC-MS/MS Conditions for Amino Acids Derivatized Using (R)-BiAC

In a previous study, we developed a method for simultaneously determining ᴅ,ʟ-amino acids using the chiral derivatizing agent (*R*)-BiAC. Good enantiomer separation, selectivity, and sensitivity were achieved using the reagent. Several types of real sample were analyzed using the method. Traditional Japanese Kurozu vinegar samples and lactic acid bacteria beverage samples were analyzed using the method, but the method has not previously been used to analyze human urine [26]. The presence of some ᴅ-amino acids in human urine can indicate kidney diseases, so it is important to develop a method for simultaneously determining chiral amino acids in urine with good sensitivity, throughput, and selectivity. The method was modified to avoid interferences caused by coexisting compounds in human urine.

The HPLC separation conditions were optimized. The column length was changed from 50 to 75 mm and the analytical time was lengthened from 14.5 to 20 min to achieve good separation between the analytes and interfering compounds. The mobile phase gradient program was also modified. A D,L-amino acid standard chromatogram obtained using the optimized conditions is shown in Figure 2. The D-alloIle and D-alloThr isomers were analyzed as counterparts of the L-isomers. Because of the characteristics of the derivatizing agent, the D-amino acid eluted faster than the L-amino acid for each analyte. This made it easier to quantify the D-amino acids, which are generally found at low concentrations in biological samples, because it avoided peak-tailing of the higher concentration L-amino acid affecting the D-amino acid peak.

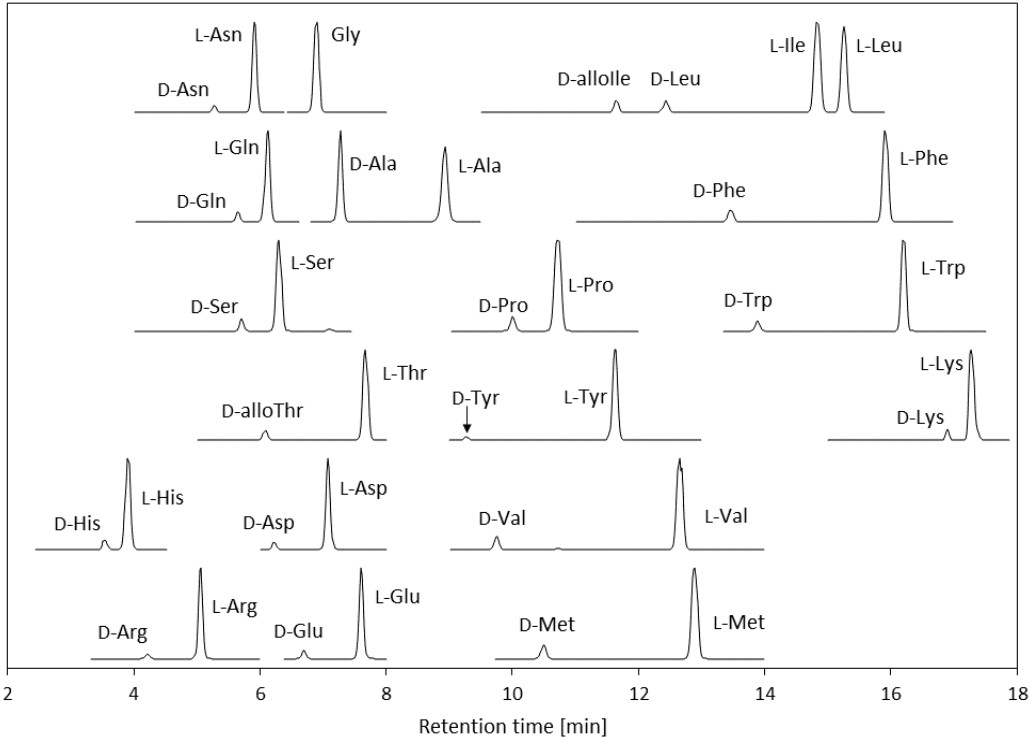

**Figure 2.** Chromatograms of the D-amino acids (1 μM), and L-amino acids and glycine (10 μM) in standard 6 (the abbreviations are defined in the text).

When performing MS analysis, the ion signal is prone to being suppressed or enhanced due to matrix effects [29]. Isotope-labeled ISs were used for all of the analytes, including the D-amino acids, to ensure that the analysis was accurate. For the method, D-amino acid ISs were prepared by racemizing the corresponding commercially available L-isomers using a previously published method [26] and a patented method [28]. The ISs were added at the beginning of the sample preparation procedure to improve the reproducibility and linearity of the method by correcting errors caused by the sample preparation procedure, including during sample transfer and derivatization. [$^{13}$C] and [$^{15}$N] uniformly labeled ISs were used to ensure that the peaks were corrected as effectively as possible because the [$^{13}$C] and [$^{15}$N]-labeled IS and analyte peaks occurred at the same elution times. Deuterium-labeled ISs were not used because some deuterium-labeled compounds elute at different retention times to their unlabeled analogs [30,31]. The IS peaks also made it easy to distinguish the analytes peaks from interfering peaks.

The MS conditions were also optimized. The D- and L-amino acid concentrations in urine samples are very different and the concentrations of different amino acids are very different. Therefore, it was difficult to determine all analytes under the same conditions because the detector would be saturated for some analytes, but not sensitive enough to detect others. The sensitivities for L-amino acids and several D-amino acids that are abundant in urine (D-Ala and D-Ser) were decreased by selecting isotopologs

(m + 1) as precursor ions to allow all of the analytes to be analyzed in one run [32]. The instrument was very sensitive to L-Gln and L-Lys, which are found at high concentrations in urine, so the (m + 2) ions of L-Gln and L-Lys were used. Divalent ions were used as precursor ions for D,L-Arg and D-Lys because the MS was more sensitive to the divalent ions than the monovalent ions of these analytes. In contrast, monovalent ions were used as precursors of L-Lys. An artifact peak eluted just after the L-Lys peak when a divalent ion transition (*m/z* 368.4) was used (Figure 3). This artifact was only detected when the *m/z* 368.4 > 295.1 transition was monitored. The optimized transitions and analytical parameters are shown in Table A1.

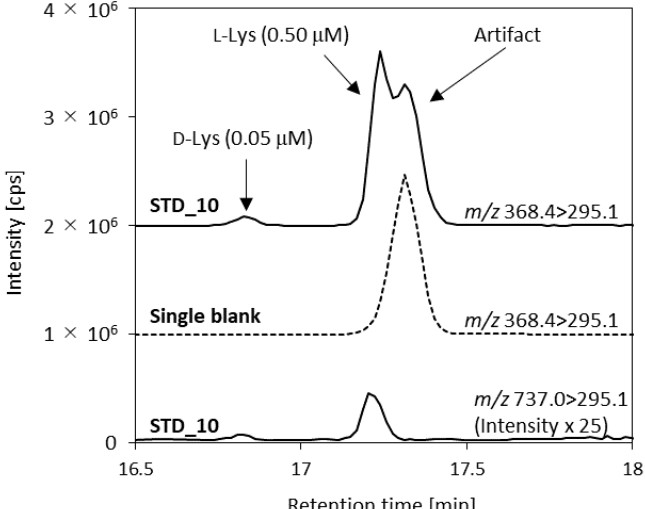

**Figure 3.** Standard 10 (STD_10) and blank chromatograms of D,L-lysine (Lys) showing the artifact that overlapped the L-Lys peak when the *m/z* 368.4 > 295.1 transition was monitored.

*3.2. Method Validation*

The developed analytical method was validated using pooled human urine sample. A D,L-amino acid chromatogram of the urine sample is shown in Figure 4. Good enantiomer or diastereomer separations were achieved under the optimized conditions. Amino acid enantiomers are endogenous metabolites that are found in human urine; therefore, standard working solutions were used as calibration standards.

3.2.1. Linearity and Quantification Ranges

The linearity of the method was assessed by analyzing the working standard solutions (STD-1–STD-12). The linear range for an amino acid was defined as the concentration range over which the specified acceptance criteria were met. The criteria were that the accuracy was ±15% or less (±20% or less for the lower quantification limit) and the precision (relative standard deviation (RSD)) (*n* = 5) was ≤15% (20% for the lower quantification limit). The correlation coefficients (r) for all of the analytes were >0.995 (data not shown). The quantification ranges for the analytes are shown in Table 1. The method was optimized for analyzing urine, and a broad quantification range was set because of the analyte concentrations found in the samples. The sensitivities for D-Ser and D-Ala were effectively decreased by using the isotopologs (m + 1) as precursor ions. The upper quantification limits for D-Ser and D-Ala were both 50 µM. In the same way, the (m + 2) isotopologs were used to decrease the sensitivities for L-Gln and L-Lys, and the upper quantification limits were both 500 µM. For the other L-amino acids, the isotopolog (m + 1) ions were used to ensure that the quantification ranges covered the actual concentrations in the urine samples.

### 3.2.2. Precision Results, Accuracies, and Recoveries

Aliquots of the pooled urine samples were spiked with standard D,L-amino acid mixtures and analyzed to evaluate intraday precision and accuracy for real human urine samples. Samples spiked at three spike concentrations (LQC, MQC, and HQC) were evaluated. To evaluate all analytes simultaneously, the LQC, MQC, and HQC concentrations of each amino acid were selected to reflect the actual concentrations of the analytes in the urine sample. Unspiked pooled urine samples and pooled urine samples spiked at the LQC, MQC, and HQC concentrations were each analyzed five times, and the RSDs and accuracies were calculated. The results are shown in Table 1. Several amino acids in the HQC-spiked sample could not be determined because the spiked concentrations caused the total concentrations to become outside of the quantification range.

**Table 1.** Quantification ranges, urine concentrations, spiked concentrations, intraday precision values, and recoveries for the analytes using the optimized method.

| Analyte | Range [μM] | Pooled Urine [1] [μM] | Spiked Concentration [μM] | | | Recovery (n = 5) | | | | | |
|---|---|---|---|---|---|---|---|---|---|---|---|
| | | | | | | RSD [%] | | | Accuracy [%] | | |
| | | | LQC | MQC | HQC | LQC | MQC | HQC | LQC | MQC | HQC |
| D-Ala | 0.025–50.0 | 3.85 | 1.25 | 5.00 | 12.5 | 3.2 | 1.2 | 1.5 | 93 | 92 | 93 |
| D-Arg | 0.025–50.0 | 0.510 | 0.250 | 1.25 | 5.00 | 4.6 | 2.8 | 2.3 | 92 | 97 | 101 |
| D-Asn | 0.010–10.0 | 1.32 | 0.250 | 1.25 | 5.00 | 1.2 | 3.6 | 2.1 | 107 | 103 | 96 |
| D-Asp | 0.025–25.0 | BLOQ [2] | 0.250 | 1.25 | 5.00 | 5.6 | 3.1 | 2.2 | 106 | 104 | 104 |
| D-Gln | 0.025–10.0 | 0.306 | 0.250 | 1.25 | 5.00 | 2.0 | 3.7 | 2.7 | 109 | 102 | 103 |
| D-Glu | 0.025–25.0 | 0.197 | 0.250 | 1.25 | 5.00 | 5.9 | 1.7 | 1.8 | 125 | 108 | 106 |
| D-His | 0.010–10.0 | 0.319 | 0.250 | 1.25 | 5.00 | 2.2 | 1.7 | 3.7 | 95 | 98 | 99 |
| D-alloIle | 0.010–5.00 | 0.064 | 0.250 | 1.25 | 5.00 | 1.7 | 2.8 | —[3] | 101 | 100 | —[3] |
| D-Leu | 0.010–5.00 | 0.072 | 0.250 | 1.25 | 5.00 | 0.6 | 1.4 | —[3] | 105 | 103 | —[3] |
| D-Lys | 0.010–5.00 | 0.484 | 0.250 | 1.25 | 5.00 | 1.6 | 2.2 | —[3] | 110 | 104 | —[3] |
| D-Met | 0.010–5.00 | 0.148 | 0.250 | 1.25 | 5.00 | 3.9 | 2.6 | —[3] | 97 | 98 | —[3] |
| D-Phe | 0.010–5.00 | 0.071 | 0.250 | 1.25 | 5.00 | 4.6 | 1.6 | —[3] | 104 | 102 | —[3] |
| D-Pro | 0.010–5.00 | BLOQ [2] | 0.250 | 1.25 | 5.00 | 2.5 | 3.2 | —[3] | 108 | 106 | —[3] |
| D-Ser | 0.100–50.0 | 9.62 | 1.25 | 5.00 | 12.5 | 2.8 | 0.6 | 4.5 | 140 | 105 | 110 |
| D-alloThr | 0.010–10.0 | 0.778 | 0.250 | 1.25 | 5.00 | 3.3 | 1.6 | 2.2 | 101 | 101 | 97 |
| D-Trp | 0.010–10.0 | BLOQ [2] | 0.250 | 1.25 | 5.00 | 2.5 | 1.2 | 3.6 | 98 | 103 | 107 |
| D-Tyr | 0.010–10.0 | 0.024 | 0.250 | 1.25 | 5.00 | 2.8 | 4.7 | 3.4 | 99 | 94 | 95 |
| D-Val | 0.010–5.00 | 0.051 | 0.250 | 1.25 | 5.00 | 1.0 | 2.6 | —[3] | 103 | 101 | —[3] |
| Gly | 0.250–500 | 132 | 12.5 | 50.0 | 125 | 3.6 [4] | 2.2 | 1.4 | 111 [4] | 101 | 102 |
| L-Ala | 0.250–500 | 25.5 | 12.5 | 50.0 | 125 | 1.6 [4] | 1.7 | 2.5 | 112 | 98 [4] | 103 |
| L-Arg | 0.250–500 | 1.21 | 2.50 | 12.5 | 50.0 | 4.0 | 9.6 | 1.9 | 102 | 107 | 102 |
| L-Asn | 0.250–500 | 7.55 | 2.50 | 12.5 | 50.0 | 2.5 | 4.3 | 1.5 | 99 | 104 | 102 |
| L-Asp | 0.500–500 | BLOQ [2] | 2.50 | 12.5 | 50.0 | 3.0 | 0.7 [4] | 2.9 | 111 | 110 [4] | 103 |
| L-Gln | 0.500–500 | 43.0 | 12.5 | 50.0 | 125 | 3.0 | 2.2 | 3.5 | 84 | 80 | 83 |
| L-Glu | 0.500–500 | 0.816 | 2.50 | 12.5 | 50.0 | 1.9 | 1.6 [4] | 1.5 | 106 | 104 [4] | 103 |
| L-His | 0.250–500 | 74.4 | 12.5 | 50.0 | 125 | 2.7 | 1.5 | 1.1 | 98 | 96 | 97 |
| L-Ile | 0.250–100 | 0.964 | 2.50 | 12.5 | 50.0 | 5.2 | 4.5 | 1.4 | 104 | 106 | 96 |
| L-Leu | 0.250–100 | 2.50 | 2.50 | 12.5 | 50.0 | 1.6 | 5.4 | 0.7 | 103 | 111 | 101 |
| L-Lys | 0.500–500 | 27.5 | 12.5 | 50.0 | 125 | 9.2 | 4.9 | 2.9 | 109 | 96 | 95 |
| L-Met | 0.250–100 | 0.641 | 2.50 | 12.5 | 50.0 | 1.9 | 2.4 | 1.0 | 99 | 101 | 101 |
| L-Phe | 0.250–50.0 | 3.96 | 2.50 | 12.5 | 50.0 | 1.6 | 3.8 | —[3] | 98 | 100 | —[3] |
| L-Pro | 0.250–100 | 0.501 | 2.50 | 12.5 | 50.0 | 2.2 | 7.9 | 2.5 | 107 | 112 | 100 |
| L-Ser | 0.250–500 | 20.8 | 12.5 | 50.0 | 125 | 1.8 [4] | 2.0 | 1.0 | 101 [4] | 99 | 101 |
| L-Thr | 0.250–250 | 14.4 | 12.5 | 50.0 | 125 | 6.6 | 2.2 | 0.9 | 108 | 95 | 93 |
| L-Trp | 0.250–100 | 6.94 | 2.50 | 12.5 | 50.0 | 1.4 | 2.1 | 2.2 | 97 | 109 | 106 |
| L-Tyr | 0.250–250 | 8.42 | 2.50 | 12.5 | 50.0 | 1.4 | 3.2 | 2.3 | 98 | 108 | 105 |
| L-Val | 0.250–100 | 2.92 | 2.50 | 12.5 | 50.0 | 1.5 | 6.5 | 2.5 | 101 | 106 | 99 |

[1] Value found after the samples were diluted by a factor of ten. [2] BLOQ below the lower limit of quantification. [3] Outside of the quantification range. [4] Calculated for four samples because one sample result was an outlier. The abbreviations are defined in the text.

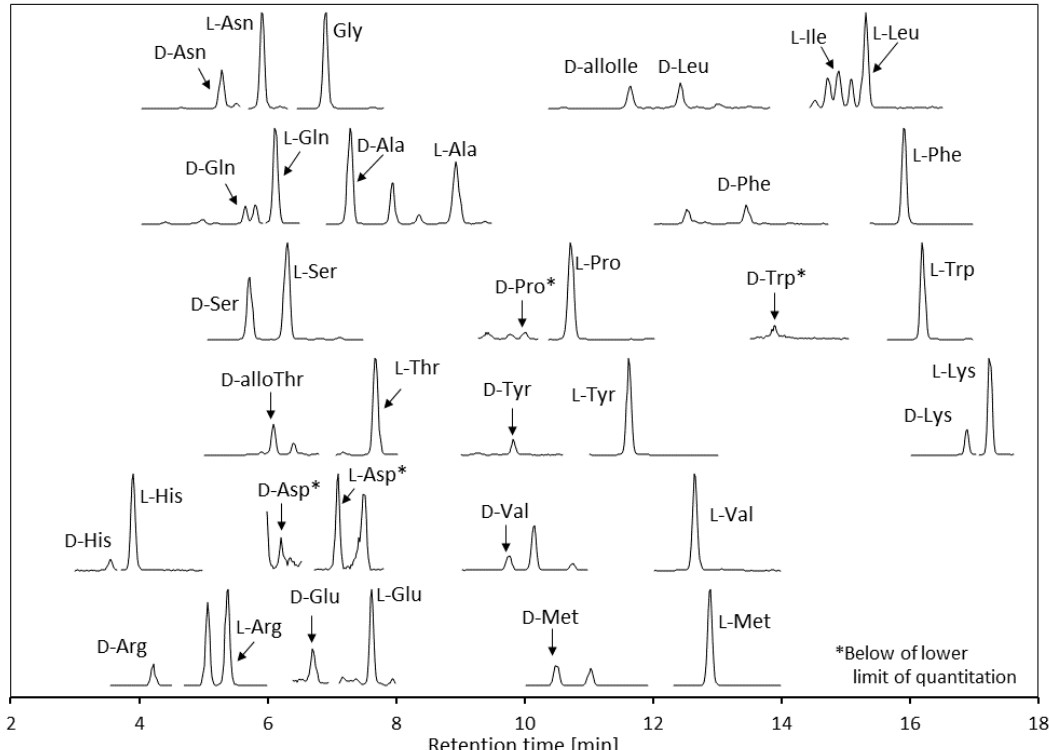

**Figure 4.** Chromatograms of the D,L-amino acids and glycine (Gly) for the pooled human urine sample (the abbreviations are defined in the text).

Good RSDs (<5.9%) were found for all of the D-amino acids. Good accuracies (92–110%) were found except for D-Ser in the LQC sample. The urine sample contained a relatively high D-Ser concentration (~10 μM), so the LQC spiked concentration (1.25 μM) was too low for the recovery test to give a reliable accuracy result. Good RSDs (<9.6%) and accuracies (93–113%) were found for the L-amino acids and Gly. The Gly, L-Asp, L-Glu, and L-Ser RSDs and accuracies were <3.6% and 101–111%, respectively. L-Gln had moderate accuracies of 80–84%. The results indicated that using the ISs for all of the compounds including the D-amino acids gave good RSDs, accuracies, and recoveries.

*3.3. Verifying the Analytical Results Using a Chirality-Switching Method Using (S)-BiAC*

Determining D-amino acids in biological samples generally suffers from interferences caused by matrix compounds because of the low D-amino acid concentrations, so it is important to verify the D-amino acid peaks. The chirality-switching method is sometimes used to confirm analytical results. The retention times of D- and L- amino acids can be switched by changing the chirality of the chiral selector [23,33]. In this study, the chiral selector was the axially chiral moiety in the derivatizing agent BiAC. Using (S)-BiAC as the derivatizing agent switched the D- and L-amino acid derivative retention times. If interfering compounds overlapped with an analyte peak, the results for the (R)- and (S)-BiAC derivatives would have changed. Analyses were performed using (S)-BiAC using the same standard solutions, ISs, and pooled urine sample. Calibration curves were constructed, then three aliquots of the pooled urine sample were analyzed. When analyzing D-Lys derivatized using (S)-BiAC, the MS/MS transition of L-Lys was used to avoid the artifact peak mentioned in Section 3.1 and shown in Figure 2. The results are shown in Table 2.

**Table 2.** Verification of the amino acid concentrations determined using (*R*)-4-nitrophenyl-*N*-[2-(diethylamino)-6,6-dimethyl-[1,1-biphenyl]-2-yl]carbamate hydrochloride ((*R*)-BiAC) and using the chirality-switching method with (*S*)-BiAC.

| Analyte | Concentration of Pooled Urine [μM] | | Ratio (*S*)-BiAC/(*R*)-BiAC |
|---|---|---|---|
| | (*R*)-BiAC [1] | (*S*)-BiAC [2] | |
| D-Ala | 3.85 | 4.17 | 108% |
| D-Arg | 0.510 | 0.497 | 97% |
| D-Asn | 1.32 | 1.38 | 105% |
| D-Asp | BLOQ [3] | 0.025 | — |
| D-Gln | 0.306 | 0.751 | 246% |
| D-Glu | 0.197 | 0.435 | 221% |
| D-His | 0.319 | 0.276 | 87% |
| D-alloIle | 0.064 | 0.043 | 67% |
| D-Leu | 0.072 | 0.791 | 1104% |
| D-Lys | 0.484 | 0.539 | 111% |
| D-Met | 0.148 | 0.149 | 101% |
| D-Phe | 0.071 | 0.077 | 109% |
| D-Pro | BLOQ [3] | BLOQ [3] | — |
| D-Ser | 9.62 | 9.74 | 101% |
| D-alloThr | 0.778 | 0.766 | 99% |
| D-Trp | BLOQ [3] | BLOQ [3] | — |
| D-Tyr | 0.024 | 0.036 | 150% |
| D-Val | 0.051 | 0.048 | 94% |
| Gly | 132 | 129 | 97% |
| L-Ala | 25.5 | 25.2 | 99% |
| L-Arg | 1.21 | 1.13 | 94% |
| L-Asn | 7.55 | 7.84 | 104% |
| L-Asp | BLOQ [3] | BLOQ [3] | — |
| L-Gln | 43.0 | 46.3 | 108% |
| L-Glu | 0.816 | 0.809 | 99% |
| L-His | 74.4 | 74.8 | 100% |
| L-Ile | 0.964 | 0.813 | 84% |
| L-Leu | 2.50 | 2.46 | 98% |
| L-Lys | 27.5 | 26.3 | 96% |
| L-Met | 0.641 | 0.666 | 104% |
| L-Phe | 3.96 | 3.84 | 97% |
| L-Pro | 0.501 | 0.516 | 103% |
| L-Ser | 20.8 | 19.8 | 95% |
| L-Thr | 14.4 | 14.2 | 99% |
| L-Trp | 6.94 | 6.84 | 98% |
| L-Tyr | 8.42 | 8.56 | 102% |
| L-Val | 2.92 | 2.90 | 99% |

[1] Concentration in urine after dilution by a factor of ten, mean for five analyses. [2] Concentration in urine after dilution by a factor of ten, mean for three analyses. [3] BLOQ below the lower limit of quantification. The abbreviations are defined in the text.

The analytical results for the analytes derivatized using (*R*)- and (*S*)-BiAC were very similar, and most of the (*S*)-BiAC/(*R*)-BiAC ratios were 84–111%. This indicated that the concentrations determined using (*R*)-BiAC were correct and were not affected by interfering compounds. The D-Trp and D-Pro concentrations determined using both (*S*)- and (*R*)-BiAC were below the lower quantification limit. The D-Asp (*S*)-BiAC/(*R*)-BiAC ratio could not be determined because the D-Asp concentration was low and could not be determined using (*R*)-BiAC. The D-Asp concentration determined using (*S*)-BiAC was also close to the lower quantification limit. The D-Gln, D-Glu, D-Leu, and D-Tyr concentrations determined using (*S*)-BiAC were >150% higher than the concentrations determined using (*R*)-BiAC. The concentrations determined using (*S*)-BiAC may have been inaccurate because of overlapping interfering peaks.

Some of the amino acids had diastereomers. The D-alloIle and D-alloThr retention times switched with the L-alloIle and L-alloThr retention times. The D-alloThr and D-Thr derivatized with (*R*)-BiAC peaks could not be separated, but the L-alloThr and L-Thr peaks could be separated (Figure 5a). The effect of D-Thr on the D-alloThr concentration could therefore be assessed using the chiral switching method. The concentrations of D-alloThr plus D-Thr determined using (*R*)- BiAC and of D-alloThr determined using (*S*)-BiAC were similar, so we concluded that D-alloThr was the main counterpart isomer of L-Thr.

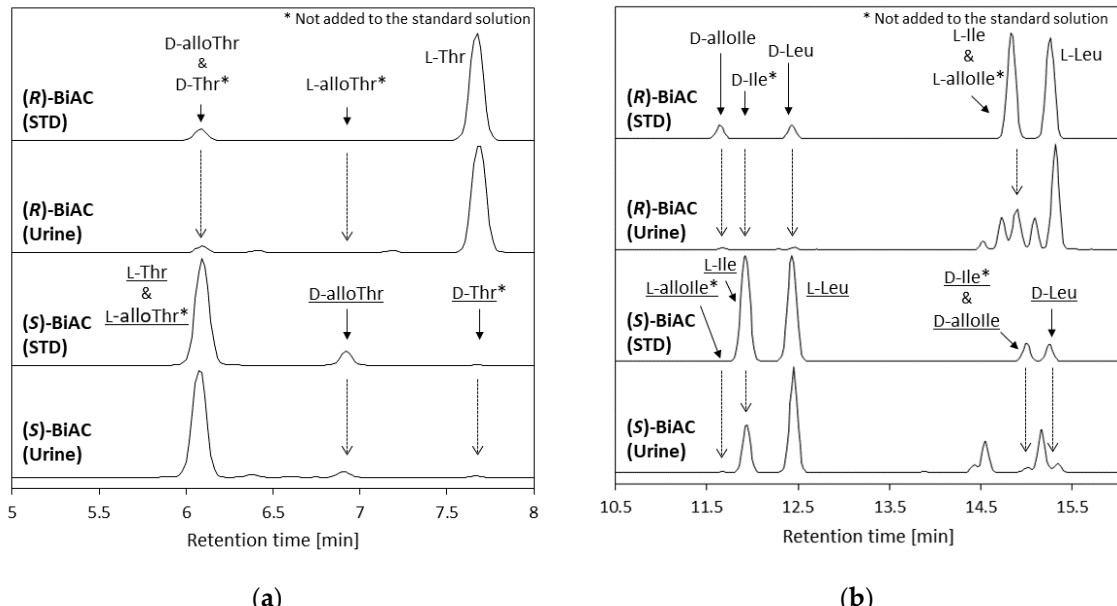

(**a**)  (**b**)

**Figure 5.** Chromatograms of standards and pooled urine samples derivatized using (*R*)-4-nitrophenyl-*N*-[2-(diethylamino)-6,6-dimethyl-[1,1-biphenyl]-2-yl]carbamate hydrochloride (BiAC) or (*S*)-BiAC. (**a**) Isomers of threonine (Thr). (**b**) Isomers of isoleucine (Ile) and leucine (Leu).

In contrast, D-alloIle and D-Ile could be separated even using (*R*)-BiAC, and the D-Ile concentration was very low. Therefore, D-alloIle was the main component of L-Ile. However, chiral-switching from (*R*)-BiAC to (*S*)-BiAC decreased the D-alloIle concentration, and the (*S*)-BiAC/(*R*)-BiAC ratio was 67%. The D-alloIle peak may therefore have been affected by some interfering compounds.

### 3.4. Human Urine Analysis

Simultaneous analysis of D,L-amino acids in human urine was performed using the optimized method. The amino acid concentrations in urine samples from five healthy volunteers are shown in Table 3. The actual concentrations in the urine samples (before the samples were diluted by a factor of ten) are shown.

Measurable concentrations of many D-amino acids were found. The concentration ranges and mean contributions of the D-amino acid to the total amino acid concentration (% D) were 89.1–246 μM and 37% D for D-Ser, 17.4–59.4 μM and 12% D for D-Ala, 11.1–24.0 μM and 17% D for D-Asn, 7.19–16.8 μM and 9.5% D for D-alloThr, 3.76–9.77 μM and 2.2% D for D-Lys, and 3.51–6.28 μM and 29% D for D-Arg. The D-Gln, D-Glu, D-His, D-Lys, D-Leu, D-Met, D-Phe, D-Trp, D-Tyr, and D-Val concentrations were lower, but quantifiable. The concentrations of all of the D-amino acids except D-Gln, D-Glu, D-alloIle, and D-Leu were verified using the chirality-switching method using (*S*)-BiAC.

The concentrations of D-amino acids in human urine have been reported in several previous publications. Armstrong et al. analyzed D-Phe, D-Trp, and D-Tyr [10], Bruckner et al. analyzed D-Ala, D-Phe, D-Ser, D-Tyr, and D-Val [11], Lorenzo et al. analyzed Ala, Val, Thr, Ser, Leu, Asp+Asn, Glu+Gln, Met, Phe, Tyr, and Lys (but only reported % D values) [34], Ishii et al. analyzed D-Ala, D-Asp, D-Glu, D-Leu, D-Pro, and D-Ser [21], and Hesaka et al. analyzed D-Asn, D-Ser, D-Gln, D-alloThr,

D-Thr, and D-Ala [17]. D-Arg and D-His concentrations could be determined in this study, but were not determined in a previous study. The % D values found in this study were similar to the % D values found in previous studies, but the D-amino acid concentrations were somewhat different, so the concentration differences may have been caused by real differences in the concentrations in the urine. In pharmacokinetic studies, the concentrations of compounds of interest in urine are corrected to the creatinine concentration. However, the creatinine concentration is affected by kidney diseases, so the % D values are promising biomarkers for urine.

**Table 3.** Quantitation results for the D,L-amino acids in human urine.

| Analyte | Concentration of Individual Urine Sample [1] [µM] | | | | | Average (n = 5) [µM] | % D [5] |
|---|---|---|---|---|---|---|---|
| | #1 | #2 | #3 | #4 | #5 | | |
| D-Ala | 19.2 | 59.4 | 21.9 | 17.4 | 57.7 | 35.1 | 12 |
| D-Arg | 6.28 | 5.83 | 4.27 | 3.51 | 5.42 | 5.06 | 29 |
| D-Asn | 24.0 | 13.3 | 11.1 | 12.9 | 14.3 | 15.1 | 17 |
| D-Asp | 0.548 | BLOQ [2] | 0.257 | BLOQ [2] | BLOQ [2] | (0.403) [3] | – |
| D-Gln | 5.13 | 2.74 | 2.51 | 3.35 | 3.59 | 3.46 | 0.8 |
| D-Glu | 3.16 | 2.53 | 1.43 | 1.59 | 2.56 | 2.25 | 22 |
| D-His | 4.64 | 2.50 | 4.46 | 3.43 | 2.40 | 3.49 | 0.5 |
| D-alloIle | 0.681 | 0.307 | 0.310 | 0.976 | 0.847 | 0.624 [4] | 5.3 [4] |
| D-Leu | 0.524 | 0.451 | 0.326 | 0.987 | 0.864 | 0.630 | 2.3 |
| D-Lys | 9.77 | 3.76 | 3.89 | 5.27 | 5.87 | 5.71 | 2.2 |
| D-Met | 0.383 | 6.517 | 0.191 | 0.273 | 0.338 | 1.54 | 16 |
| D-Phe | 0.848 | 0.538 | 0.400 | 0.706 | 1.03 | 0.705 | 1.5 |
| D-Pro | BLOQ [2] | BLOQ [2] | BLOQ [2] | BLOQ [2] | BLOQ [2] | – | – |
| D-Ser | 246 | 94.2 | 93.5 | 105 | 89.1 | 126 | 37 |
| D-alloThr | 16.8 | 9.52 | 7.19 | 7.54 | 8.00 | 9.80 | 9.5 |
| D-Trp | 0.106 | BLOQ [2] | BLOQ [2] | 0.124 | BLOQ [2] | (0.115) [3] | 0.2 |
| D-Tyr | 0.298 | 0.225 | 0.109 | 0.191 | 0.338 | 0.232 | 0.2 |
| D-Val | 0.829 | 0.514 | 0.366 | 0.459 | 0.638 | 0.561 | 1.6 |
| Gly | 2,120 | 761 | 2,470 | 1,350 | 685 | 1,480 | – |
| L-Ala | 300 | 166 | 234 | 278 | 320 | 260 | – |
| L-Arg | 13.9 | 13.2 | 8.98 | 15.0 | 9.44 | 12.1 | – |
| L-Asn | 61.8 | 56.9 | 87.5 | 91.7 | 63.2 | 72.2 | – |
| L-Asp | BLOQ [2] | BLOQ [2] | BLOQ [2] | BLOQ [2] | BLOQ [2] | – | – |
| L-Gln | 553 | 290 | 475 | 511 | 371 | 440 | – |
| L-Glu | 12.1 | 5.45 | 6.68 | 11.3 | 5.62 | 8.23 | – |
| L-His | 966 | 584 | 763 | 806 | 695 | 763 | – |
| L-Ile | 17.5 | 7.51 | 8.09 | 10.8 | 11.4 | 11.1 | – |
| L-Leu | 40.7 | 19.2 | 21.0 | 29.3 | 26.2 | 27.3 | – |
| L-Lys | 154 | 201 | 184 | 338 | 407 | 257 | – |
| L-Met | 14.0 | 5.49 | 7.53 | 9.28 | 4.96 | 8.25 | – |
| L-Phe | 74.4 | 37.1 | 34.2 | 43.2 | 39.9 | 45.7 | – |
| L-Pro | 6.14 | 3.69 | 4.93 | 6.67 | 4.14 | 5.11 | – |
| L-Ser | 227 | 111 | 314 | 287 | 110 | 210 | – |
| L-Thr | 173 | 94.6 | 176 | 196 | 96.3 | 147 | – |
| L-Trp | 91.4 | 51.1 | 69.5 | 69.2 | 78.1 | 71.9 | – |
| L-Tyr | 126 | 82.1 | 73.5 | 89.6 | 95.2 | 93.4 | – |
| L-Val | 53.4 | 23.7 | 29.9 | 28.5 | 31.8 | 33.5 | – |

[1] Actual concentration. [2] BLOQ below the lower limit of quantification. [3] Mean for the two samples that could be quantified. [4] Possibly inaccurate because of coelution with interfering compounds. [5] Calculated using the mean concentrations. %D = D-amino acid concentration/(D-amino acid concentration + L-amino acid concentration) × 100. The abbreviations are defined in the text.

The method allowed proteinogenic D,L-amino acids to be successfully simultaneously analyzed with high degrees of sensitivity and selectivity. The chiral-switching method verified the D-amino acid concentrations. The method gives reliable results, particularly for samples containing large amounts of

interfering compounds. The method is expected to be able to be used for other challenging biological samples such as tissues and feces.

## 4. Conclusions

A method for simultaneously determining 18 chiral proteinogenic amino acids and Gly in human urine samples using the axially chiral derivatizing agent (*R*)-BiAC was developed. The optimized conditions allowed 37 analytes to be separated in 20 min. Using appropriate MS transitions allowed a broad dynamic range to be covered. Stable-isotope-labeled ISs for all analytes were used, and good linearities, precision results, and recoveries were obtained. The analytical method was verified by performing the analysis using (*S*)-BiAC, which has the opposite chirality to (*R*)-BiAC. Using (*S*)-BiAC caused the retention times of the derivatized D- and L-amino acids to switch, allowing the concentrations determined using (*R*)-BiAC to be checked. The results obtained using (*S*)-BiAC were consistent with the results obtained using (*R*)-BiAC except for several amino acids such as D-alloIle. The method allowed five human urine volunteer samples to be successfully analyzed. The method was validated for proteinogenic amino acids, but may be able to be used for nonproteinogenic amino acids. Combined with the chirality-switching method, the method is expected to be able to be used to screen biomarkers and elucidate the mechanisms involved in diseases.

**Author Contributions:** Conceptualization and methodology, M.H. and S.K.; validation and investigation, M.H.; data curation, M.H.; writing—original draft preparation, M.H.; writing—review and editing, S.K. and K.S.; supervision, H.M. and K.S. All authors have read and agreed to the published version of the manuscript.

**Funding:** This research received no external funding.

**Acknowledgments:** Akira Nakayama and Daigo Iwahata are gratefully acknowledged for helpful discussions. We are grateful for the technical support provided by Satomi Kobayashi.

**Conflicts of Interest:** The authors declare no conflict of interest.

## Appendix A

**Table A1.** Selected precursor ions and analytical parameters.

| Analyte | Precursor Ion [*m/z*] | | | DP | CE | CXP |
|---|---|---|---|---|---|---|
| | Analyte (D-isomer) | Analyte (L-isomer) | IS [4] | | | |
| Ala | 385.1 [1] | 385.1 [1] | 388.1 | 136 | 29 | 10 |
| Arg | 235.2 | 235.7 [1] | 240.0 | 46 | 17 | 10 |
| Asn | 427.1 | 428.0 [1] | 433.0 | 136 | 29 | 10 |
| Asp | 428.2 | 429.2 [1] | 433.1 | 66 | 35 | 10 |
| Gln | 441.0 | 443.0 [2] | 448.1 | 136 | 29 | 10 |
| Glu | 442.1 | 443.0 [1] | 448.0 | 66 | 35 | 10 |
| His | 450.1 | 451.1 [1] | 459.1 | 66 | 23 | 16 |
| Ile | 426.0 [3] | 427.0 [1] | 433.1 | 136 | 29 | 10 |
| Leu | 426.0 | 427.0 [1] | 433.1 | 136 | 29 | 10 |
| Lys (2+) | 368.4 | – | 372.4 | 46 | 29 | 36 |
| Lys (1+) | – | 737.0 [2] | 743.0 | 106 | 39 | 18 |
| Met | 444.0 | 445.0 [1] | 450.0 | 136 | 29 | 10 |
| Phe | 460.1 | 461.1 [1] | 470.0 | 26 | 33 | 8 |
| Pro | 410.2 | 411.2 [1] | 416.0 | 46 | 31 | 8 |
| Ser | 401.0 [1] | 401.0 [1] | 404.0 | 136 | 29 | 10 |
| Thr | 414.0 [3] | 415.0 [1] | 419.0 | 136 | 29 | 10 |
| Trp | 499.0 | 500.0 [1] | 512.0 | 26 | 33 | 8 |
| Tyr | 476.0 | 477.0 [1] | 486.0 | 26 | 33 | 8 |
| Val | 412.0 | 413.0 [1] | 418.0 | 136 | 29 | 10 |
| Gly | 371.0 [1] | | 373.0 | 136 | 29 | 10 |

[1] Isotopolog (m + 1) was used as a precursor. [2] Isotopolog (m + 2) was used as a precursor. [3] The D-alloIle isomer was analyzed. [4] The same ions were monitored for both the D- and L-isomers. DP declustering potential, CE collision energy, CXP collision cell exit potential.

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
