# Peer review of "Simultaneous Analysis of d,l-Amino Acids in Human Urine Using a Chirality-Switchable Biaryl Axial Tag and Liquid Chromatography Electrospray Ionization Tandem Mass Spectrometry"

_symmetry, doi:10.3390/sym12060913_

Round 1

Reviewer 1 Report

 Last decade is characterized by dynamic development of instrumental methods, that results in advance and diversity of applied analytical procedures for separation and analysis of chiral compounds. The article describes a method for the simultaneous determination og 18 chiral proteinogenic amino acids  in human urine samples using an axially chiral derivatizing agent. The optimized conditions allowed 37 analytes to be separated in 20 minutes which is excellent. The method was validated for proteinogenic amino acids but may be able to be used for non-proteinogenic amino acids, and has the potential to be used to screen biomarkers and elucidate the mechanisms involved in certain diseases. The manuscript requires a minor redaction of grammatical and stylistic errors. Congratulations on this study, valuable for researchers who are involved in the analysis of chiral drugs.  Success in future researches.

Author Response

Thank you for your comment. The redaction of the grammatical and stylistic errors (style of the references) has been conducted.

Reviewer 2 Report

This manuscript optimized a method for the simultaneous detection of D and L-amino acids in human urine using the chiral derivatizing agent (R)-BiAC, and the method was cross-verified using (S)-BiAC. I found the method well described and the results very interesting.

Here are some minor issues that need to be addressed.

  1. Structure (a) in Figure 1 has an extra thick bond next to the aromatic -NO2
  2. In the Method section 2.6 (page 4), it was not clear to me how much equivalent urine was eventually injected onto the column. In page 4 line 139, it says “the sample mixture or a 20 µL aliquot of a standard (for calibration) was then mixed with 20 µL of the IS solution and 40 µL of acetonitrile.” Was the volume of “sample mixture” also 20 uL? Page 10 line 308 mentioned that “before the samples were diluted by a factor of five.” I’m not sure where the number “five” came from. In page 4 line 137, the individual urine samples were diluted by 10 times (50 µL urine mixed with 450 µL of water).
  3. Please add references to the first paragraph in 3.1, where “previous study” and “traditional Japanese Kurozu vinegar samples and lactic acid bacteria beverage samples” were analyzed. It was not clear what method was modified as was mentioned in the last sentence of this paragraph.
  4. Please change the phrase in page 5 line 182, “because by”.
  5. Figure 5 (a), third row from top, was L-Thr added or not to the standard solution? Do you mean “L-Thr & L-alloThu*”? Similarly, in Figure 5 (b) third row from top, it said “D-Ile & D-alloIle*”, but the top row was “D-Ile*” and “D-alloIle”. Was D-Ile or D-alloIle added to the standard solution?
  6. The format of journal names in Reference were not consistent. Some were full names, while others were abbreviations.

Author Response

We have made a correction of the text according to the comment.

1.Structure (a) in Figure 1 has an extra thick bond next to the aromatic -NO2

Response 1: Thank you for your comment. I am not sure which bond you are pointing to. The biphenyl structure (the left side of the structure) is twisted, but the 4-nitrophenyl group (leaving group) is not. Only the biphenyl groups use the thick bond to show their axially chiral structures. The bond may appear thicker depending on the display environment.

 2.In the Method section 2.6 (page 4), it was not clear to me how much equivalent urine was eventually injected onto the column. In page 4 line 139, it says “the sample mixture or a 20 µL aliquot of a standard (for calibration) was then mixed with 20 µL of the IS solution and 40 µL of acetonitrile.” Was the volume of “sample mixture” also 20 uL? Page 10 line 308 mentioned that “before the samples were diluted by a factor of five.” I’m not sure where the number “five” came from. In page 4 line 137, the individual urine samples were diluted by 10 times (50 µL urine mixed with 450 µL of water).

Response 2: Thank you for your comment. As you mentioned, the sample volume was not clearly described. The sample mixture was also 20 uL, and we made a correction of the text. The urine samples were diluted by 10 times, so the factor should be “ten”. Therefore, the results described in Table 3 were also corrected. The conclusion is not changed by the correction.

 3.Please add references to the first paragraph in 3.1, where “previous study” and “traditional Japanese Kurozu vinegar samples and lactic acid bacteria beverage samples” were analyzed. It was not clear what method was modified as was mentioned in the last sentence of this paragraph.

Response 3: Thank you for your comment. We added the reference here.

 4.Please change the phrase in page 5 line 182, “because by”.

Response 4: Thank you for your comment. We change the phrase to “due to”.

 5.Figure 5 (a), third row from top, was L-Thr added or not to the standard solution? Do you mean “L-Thr & L-alloThu*”? Similarly, in Figure 5 (b) third row from top, it said “D-Ile & D-alloIle*”, but the top row was “D-Ile*” and “D-alloIle”. Was D-Ile or D-alloIle added to the standard solution?

Response 5: Thank you for your comment. As you mentioned, the text shown in Figure 5 (a) should be “L-Thr & L-alloThr*”, and also in Figure 5 (b) should be “D-Ile* and L-alloIle”. In the standard solutions, L-alloThr and D-Ile were not added. The words and the position of the asterisk mark were corrected. Figure 5 is changed as well.

 6.The format of journal names in Reference were not consistent. Some were full names, while others were abbreviations.

Response 6: Thank you for your comment. We made a correction of the abbreviations of the journal names.